# The Influence of Manganese on Growth Processes of *Hordeum* L. (Poaceae) Seedlings

**DOI:** 10.3390/plants10051009

**Published:** 2021-05-19

**Authors:** Kirill Tkachenko, Irina Kosareva, Marina Frontasyeva

**Affiliations:** 1Komarov Botanical Institute of RAS (BIN), 197376 Saint-Petersburg, Russia; kigatka@gmail.com; 2The N.I. Vavilov All-Russian Institute of Plant Genetic Resources (VIR), 190000 Saint-Petersburg, Russia; irkos2004@yandex.ru; 3Sector of Neutron Activation Analysis and Applied Research, Frank Laboratory of Neutron Physics, Joint Institute for Nuclear Research, 141980 Dubna, Russia

**Keywords:** heavy metals, soil pollution, barley, collections, varieties, resistance to manganese, root length index

## Abstract

Manganese, as one of the xenobionts, belongs to the group of heavy metals, which, in high concentrations, can negatively affect the development of plants. In small concentrations, it is necessary for plants for normal growth and development. It is present in soils and is available to plants to varying degrees. In acidic soils, it often acts as a toxic element, and plants do not develop well and can even die. Screening major crops for manganese tolerance is essential. Based on the analysis of the collection of barley (*Hordeum* L., Poaceae), the N.I. Vavilov All-Russian Institute of Plant Genetic Resources (VIR) presented data that manganese-tolerant varieties and samples are concentrated in western and northern countries with a wide distribution of soils with low pH levels and high contents of mobile manganese. It follows from the diagnostic results that the maximum number of barley genotypes resistant to manganese is concentrated in Sweden, Finland, the northwestern and northern regions of the CIS countries, and the Russian Federation. In most cases, the samples tolerant to Al showed resistance to Mn as well, which is of great interest for further study of the mechanisms of plant resistance to these stressors. As a rule, samples from the northern territories—zones of distribution of acidic soils—were highly resistant. In this case, the role of the species belonging to the sample was leveled out. The highlighted areas (Scandinavia (Finland, Sweden), northern and northwestern regions of Russia, Belarus, and the Baltic countries) are sources of germplasm valuable for selection for acid resistance of barley.

## 1. Introduction

The analysis of a significant number of published works [1,2,3,4,5,6,7,8,9,10,11,12,13,14,15,16] on relative Al, heavy metals, and their influence on various processes in plants makes it possible to formulate a number of provisions on the importance of metals in plant life and, in particular, the role of manganese, including as a xenobiotic.

Among the secondary effects of pollution of natural and agricultural ecosystems is damage to terrestrial vegetation due to changes in soil characteristics, and a corresponding change in the nature of plant nutrition [4,5,6,15,17,18,19,20,21,22].

Among toxicants, heavy metals and their compounds are a significant hazard. The greatest danger is posed by labile forms, which are characterized by high biochemical activity and which accumulate in biological media.

In plant liquids and extracts, manganese is present in the form of free cationic forms and is transported in plants in the form of Mn^2+^; however, in phloem exudates, complex compounds of manganese with organic molecules are found. The lower concentration of manganese in the phloem exudate, in comparison with the leaf tissue, and the weak movement of the element in the phloem vessels cause a low content of manganese in roots, seeds, and fruits. An excess of manganese leads to oppression and even to the death of plants (exhibiting pronounced phytotoxicity). The toxicity of this element is most pronounced in acidic soddy-podzolic soils, especially at high humidity, with crust formation, and with the introduction of physiologically acidic fertilizers without neutralizing them [23].

One of the reserves for the growth of barley productivity is the increase in the acid resistance of newly created varieties. The N.I. Vavilov All-Russian Institute of Plant Genetic Resources (VIR) has developed modifications of methods for screening the gene pool of barley and other crops for resistance to H^+^, Al^3+^, and Mn^2+^ ions, which makes it possible to differentiate samples into resistance groups rather quickly and with minimal cost [23,24,25]. These modifications involve the use of an aqueous culture with variants of stress backgrounds—an excess of manganese in the form of aqueous manganese chloride with a pH of 4. The diagnostic criterion of the method is the root length index (RLI) of the seedlings. An integrated diagnostic indicator was determined that simultaneously reflected the resistance of the genotype to an excess of mobile manganese, which is the coefficient of variation of the root length indices of the seedlings cultivated under control conditions and on stress backgrounds. The authors proposed a gradation of acid resistance according to the coefficient of variation of RLI.

The aim of this work was to determine the influence of the concentration of heavy metals, using the example of manganese, on the formation of barley seedlings, which were the test objects.

The relevance of the work lies in the fact that the methods used made it possible to carry out an express assessment of the presence of pollutants in the environment; they were simpler and more accessible in comparison with physicochemical methods of determination.

To achieve this goal, we focused on the following: -to study the effect of 6-aqueous manganese chloride against the background of low pH at concentrations maximally differentiating the growth of embryonic roots of barley samples from the VIR gene pool;-to assess the possibility of using barley seedlings as a potential test object (the growth of barley seedling roots with an excess of manganese against the background of normal nutrition).

## 2. Materials and Methods

The diagnosis of plant resistance to metallotoxicity can be carried out in various ways. The analysis of the growth parameters and productivity of plants in the field is considered a direct method; it assumes the presence of soil areas with different concentrations of the studied metal, which is difficult to achieve in real conditions if there is not a testing ground. The pH of the soil solution changes significantly in the surface and subsurface horizons of the soil; the manifestation of various infections also depends on its properties. This method is expensive, time-consuming, and labor-intensive.

Indirect assessment methods include vegetation and laboratory methods. The vegetation method assumes the use of natural, acidic soil (if it is resistant to metals with a low pH); in the control variant, it is limed [26].

At present, laboratory methods for assessing plants for metallotoxicity, which are based on the use of culture in nutrient solutions and the introduction of the desired metal into the solution, have received great development and application. These screening approaches allow for avoiding “noise” and diagnosing breeding material in the early periods of plant growth and development, which increases the throughput and provides a low cost of research. In addition, this makes it possible to carry out the diagnostics of genetic diversity in vivo and to select genotypes with contrasting resistance.

Modifications of the culture method in nutrient solutions differ in technical equipment and diagnostic evaluation criteria. In the studies, the practice of growing plants in special nurseries with a mesh bottom was adopted [23,26,27,28].

The laboratory method for assessing resistance to metallotoxicity requires the use of healthy seed material, preferably one year old, and a place of reproduction with high germination and germination energy.

The degree of adverse effects on plants depends on the associated conditions (temperature, photoperiod, and light level); therefore, it was important to maintain the specified environmental conditions.

The screening of samples in an aqueous culture during the analysis of metal resistance in an acidic medium dictated the need to maintain a given pH since, during the development of seedlings, a shift towards alkaline values is observed.

Morphometric changes in the root system are one of the early negative responses of plant seedlings to an excess of hydrogen ions and available forms of metals in the nutrient solution [23,27].

Method: To determine the resistance of genotypes to an excess of Mn^2+^ at low pH values, a modification of the method based on the inhibition of growth parameters in plants exposed to this toxicant was developed. Seeds of different varieties of *Hordeum* L. (Poaceae family) from the Vavilov Institute collections (the resistance or sensitivity to manganese was tested on 385 varieties) were laid out in Petri dishes, with distilled water (10 mL) and 50 seeds of each sample added to each one. Each experiment was carried out with three repetitions. They were placed in a thermostat with a temperature of 21 °C. After pecking the seeds, the samples were laid out in cells with a mesh bottom, the bottom of which touched the surface of oxygen-enriched solutions according to the experimental scheme. According to the experiment scheme, 4.5 L of the required solution was poured into each plastic container. On the first day, the nurseries with the samples were covered with plastic wrap and immediately placed in a climatizer with specified environmental conditions: a temperature of 21 °C during the day, 18 °C at night, a photoperiod of 16 h, and illumination of 15 klx. The test character in the performed screening was an index equal to the ratio of the length of seedling roots under stressful conditions to the length of seedling roots without a stressor 

Experiment scheme: (1) Control—distilled water, pH 6.0 (according to the classical methods of seed germination, it is always set at a pH of 6.0 and Mn^2+^ of 0); (2) Stress background—four-aqueous manganese chloride (MnCl_2_ × 4H_2_O) with a concentration of 20 mg/L dissolved in distilled water; pH 4 (this is a model of acidic soils most typical for northwest Russia).

Five days after the seed placement, the maximum lengths of the roots of each seedling were measured. The root length index was calculated as the ratio of the average root length of the stress variant to the control. According to the data obtained during the statistical analysis, this indicator was the most informative for characterizing the resistance to the metallotoxicity of melilot [26].

For each group of genotypes (resistant to free soil metals (Group I) and sensitive to the metal toxicity of acidic soils (Group II)), the arithmetic means of the root lengths and leaf lengths under the influence of Mn^2+^ and H^+^ were calculated, and the error of these averages was determined. The significance of differences in root lengths and leaf lengths between the group of resistant (Group I) and sensitive (Group II) genotypes when exposed to toxicants was assessed using Student’s *t*-test.

Table 1 shows that low concentrations of manganese chloride against a background of low pH enhance the growth of roots and shoots of barley seedlings, and an increase in the content of manganese chloride decreases them against a background of low pH. The higher the concentration of MnCl_2_ × 4H_2_O mg/L, the more the growth of the aboveground and underground (roots) parts of the barley plants changed. The most negative effect was noted at a dose of 20 mg/L, as indicated by the correlation analysis data [28].

A correlation analysis was performed between the lengths of the sprouts and the roots and manganese content. The error in the correlation coefficient was calculated using a well-known formula [29]. Because the coefficient correlation serves as an estimate of its general parameter p, the null hypothesis was tested, that is, the assumptions were that, in the general population, *p* = 0. For this, t_f_ was calculated and compared with t_st_ at a significance level of 0.1% for k = *n* − 2 = 28. t_f_ is greater than or equal to t_st_.

## 3. Results and Discussion

Barley (*Hordeum* L.) is one of the oldest and most important food crops that is widespread on earth [30,31,32]. It is weakly resistant to increased soil acidity, which is associated with an underdeveloped root system of the plant and a low ability to chelate. Cultural two-rowed barley, according to the degree of reduction (or underdevelopment) of lateral sterile spikelets, is divided into two independent groups:(a)The nutantia group has relatively underdeveloped lateral spikelets, which retain the spikelet scales, as well as rather well-developed outer and inner flower films, and sometimes stamens.(b)The group deficientia has more underdeveloped lateral spikelets, of which only spikelet scales have been preserved.

The difference between the listed subspecies and groups of barley can be schematically represented as follows:A.All spikelets are fruitful, that is, all three of the spikelets sitting on the ledges of the spikelet bear the grain vulgare—multi-row barley.B.The number of fruiting spikelets on the barley ledges is different, from one to three intermedium—intermediate barley.C.Only medium spikelets are fruitful; only the middle of the three spikelets sitting on the ledges of the spikelet develops the distichum grain—two-row barley:
(a)Lateral sterile spikelets have both spikelet and flower scales, and sometimes stamens—the nutantia group.(b)Lateral sterile spikelets have only spikelet scales—the deficientia group.

Multi-row barley and the group of cultivated two-row barley are of the most practical value.

The results of Table 2 show that the informativeness of the growth of the root length (RLI) is significantly higher than the growth rates of the aerial part.

Table 2 shows the results of calculating the indices of the length of the root and leaf (sprout) of barley varieties with contrasting resistance to mobile aluminum [27]. The varieties Moskovsky 121, Polyarny 14, and Djugay are resistant to aluminum, the rest of the varieties are unstable when exposed to this metal. From the results of the table, it follows that the varieties resistant to aluminum changed the root length index when exposed to 253 mg/L of manganese to a lesser extent than unstable varieties. This may be due to the fact that natural acidic soils are often characterized not only by a high content of mobile aluminum but also by manganese, and we have the results of natural selection for acid resistance.

In barley varieties, the root length index under the influence of manganese changes significantly more than the leaf length. In this regard, this indicator was chosen for the mass laboratory screening of varieties and samples of barley for resistance to mobile manganese.

The screening of the 385 varieties in the world collection of barley (Table 3), which included samples differing in species and ecological-geographical origin, made it possible to identify samples with specific resistance to the toxic mobile metal of acidic soil Mn. Samples of various ecological and geographical origins were included: regions of the Russian Federation, Commonwealth of Independent States (CIS) countries, Western Europe, Asia, and Scandinavia.

Table 3 shows the results of a mass laboratory screening of barley samples for resistance to mobile manganese at a concentration of 253 g/L and a pH of 4.0. The table shows the number of samples studied, the average value of the root length index under the influence of manganese and pH, as well as the variability of the index.

The unequal number of samples selected in the collection from different countries was associated with the earlier assessment of aluminum resistance and a special interest in the Russian Federation and Scandinavian countries.

The variability of the index, as follows from the table, is significant. This is due to the significant variability of the manganese resistance of the samples.

The highest manganese tolerance was demonstrated by the samples and varieties from Sweden; the root growth index for them was 0.70. Samples and varieties from countries with a high prevalence of acidic soils often had a high index, which affected its average value. These are RF, Lithuania, Estonia, Belarus, Sweden, Finland, and Canada.

The variability of the root length indices ranged from 0.96 in the resistant sample to 0.11 in the unstable one. This suggests that among the genetic diversity there is a dependence of the growth of the embryonic roots of a particular sample on the manganese content in the substrate. In addition, it is possible to select samples that contrastingly react to the excess of mobile manganese, which, after additional study, may be testers for the uptake and accumulation of heavy metals from the soil. Similar results using the water culture method were obtained on 6 varieties of barley and from other researchers [16].

From the results obtained (Table 2 and Table 4), it follows that manganese-resistant varieties and samples are found in western and northern countries with a wide distribution, low pH, and mobile manganese. Similar results were observed when screening cultures for aluminum resistance [23]. It follows from the diagnostic results that the maximum number of manganese-resistant barley genotypes is concentrated in Sweden, Finland, and the northwestern and northern regions of the CIS and the Russian Federation.

The reaction of the Novichok barley variety to heavy metals in the field was carried out [33]. It was found that plants grown on contaminated soils were characterized by the highest accumulation of zinc and lead. The accumulation of heavy metals in the roots was higher than in the shoots. The data obtained by us in the study of the toxicity of manganese are quite consistent with the results of studies carried out in aquatic culture on 5 varieties of barley [16]. The increased content of manganese in the experiments of these authors led to a significant effect on the morphometric parameters of seedling roots. It was found that an increase in the dose of manganese in the growing medium of plants leads to a decrease in the length of the roots and an increase in the antioxidant activity of the enzyme superoxide dismutase; to a greater extent, this was expressed in the roots of the seedlings.

## 4. Conclusions

The screening of 385 samples and varieties of barley from the VIR collection in water culture and controlled environmental conditions with an excess content of mobile manganese against the background of control conditions (without manganese) made it possible to note a significant variability of the RLI index in the bulk fragment of the collection. The decrease in the average length of the roots of the stressor variant reached, on average, 89%, in comparison with the control in unstable samples, and reached 4% in the resistant ones.

It was found that the relative growth of the roots of barley samples was significantly inhibited under the action of a stressor (the growth of seedlings of barley samples at a Mn concentration of 253 mg/L for 5 days).

Large-scale screening of 385 barley samples showed that the value of the ratio of seedling root length obtained against a stress background (enriched with 4-aqueous manganese chloride with a pH of 4.0) to root length under control growing conditions varies significantly between samples. Thus, there is a genetic variation: the manifestation of the trait of resistance to manganese toxicity among barley genotypes.

From the world collection of barley cultivars, samples were selected, the growth of the roots of which was the least inhibited under the action of a stressor. The analysis of the literature data indicates that such samples are often resistant to other metals, in particular, aluminum.

The highest resistance to mobile manganese was shown by the varieties from Finland (Hja 81,205 (RLI = 0.94), p. Eero (0.94)); the northwest of Russia (Belogorskiy village, 0.96) and from Lithuania, Djugay village (0.93). Thus, the highest resistance to mobile manganese was shown by the varieties of the northern and northwestern territories characterized by significant areas of soddy-podzolic acid soils.

In our opinion, this indicates the natural selection of samples in these territories. This also testifies to the adequate work of the diagnostic method we have chosen.

In this study, we have shown the level of genetic variability of barley samples according to the response of root growth to an excess of mobile manganese. The selected samples can be used in breeding studies to create acid-resistant varieties of barley. Isolated samples often carry genes for aluminum resistance, as shown by the authors. The presence of the property of resistance to metallotoxicity in these samples dictates the need to study them in soil enriched with heavy metals.

## Figures and Tables

**Table 1 plants-10-01009-t001:** Influence of increasing concentrations of Mn^2+^ ions on the growth parameters of *Hordeum* L. varieties (water culture).

Variety	pH	Concentration of MnCl_2_ × 4H_2_O mg/L	Shoot Length, mm	Correlation Coefficient	t_f_	Root Length, mm	Correlation Coefficient	t_f_
Moskovsky 10	6.5	0	139 ± 8			51 ± 8		
	4.0	25	169 ± 9	−0.90 ± 0.1	8.7	96 ± 14	−0.72 ± 0.1	5.5
	4.0	63	143 ± 7			87 ± 13		
	4.0	125	112 ± 13			64 ± 11		
	4.0	189	111 ± 6			48 ± 3		
	4.0	253	89 ± 5			27 ± 3		
Nutans 88	6.5	0	164 ± 10			48 ± 4		
	4.0	25	153 ± 5	−0.82 ± 0.1	7.6	80 ± 8	−0.65 ± 0.1	4.5
	4.0	63	173 ± 13			50 ± 2		
	4.0	125	152 ± 5			51 ± 4		
	4.0	189	133 ± 4			39 ± 3		
	4.0	253	133 ± 5			31 ± 5		

**Table 2 plants-10-01009-t002:** Indices of root length and leaf length (when exposed to manganese) of *Hordeum* L. varieties with contrasting metal resistance.

Variety	Roots			Aboveground Part		
	Root Length Index	Average Root Length, mm	t_f_	Leaf Length Index	Average Leaf Length, mm	t_f_
Moskovsky 121	0.69	79 ± 2.0	2.2	0.87	138 ± 3.0	0.59
Polyarniy 14	0.67			0.97		
Djugay	0.93			0.97		
Prairie	0.38	32 ± 1.0		0.88	121 ± 3.0	
Odesskiy 100	0.28			0.93		
Donetskiy 8	0.27			0.89		

**Table 3 plants-10-01009-t003:** Indices of root length for *Hordeum* L. samples of various ecological and geographical origins.

Origin of the Group of Samples	Number of Samples, pcs.	Average Root Index	Index Variability
Russian Federation	100	0.57	0.13–0.96
Estonia	1	0.50	0.40–0.59
Latvia	2	0.67	0.50–0.85
Lithuania	2	0.79	0.65–0.93
Byelorussia	6	0.62	0.33–0.88
Moldavia	2	0.43	0.41–0.46
Ukraine	22	0.32	0.18–0.52
Uzbekistan	2	0.49	0.40–0.59
Kazakhstan	3	0.36	0.27–0.44
Sweden	114	0.70	0.39–0.87
Denmark	1	0.41	0.38–0.46
Norway	10	0.45	0.32–0.56
Finland	89	0.66	0.37–0.96
Great Britain	4	0.50	0.42–0.85
Czech	11	0.35	0.23–0.54
India	1	0.41	0.37–0.46
Canada	3	0.57	0.56–0.59
United States of America	3	0.52	0.51–0.53
Mexico	9	0.24	0.11–0.39

**Table 4 plants-10-01009-t004:** Barley *Hordeum* L. samples distinguished by resistance to excess.

VIR Catalog Number	Variety Name	Variety	Sample Origin	Root Length Index
22089	Belogorsk	pallidum ^1^ + rikotense	Russian Federation, Leningrad region	0.96
24014	Olympus	medicum ^3^	Russian Federation, Omsk region	0.83
29237	BC 793/904	nutans ^2^	Russian Federation, Omsk region	0.89
27700	Sire 2	nutans ^2^	Russian Federation, Novosibirsk region	0.81
70079	Liisa	nutans ^2^	Latvia	0.85
16928	Djugay	nutans ^2^	Lithuania	0.93
27606	Yanka	nutans ^2^	Byelorussia	0.88
23900	WW 6472	erectum	Sweden	0.84
23902	WW 6259	nutans ^2^	Sweden	0.81
23906	WW 6517	nutans ^2^	Sweden	0.82
23916	WW 6346	nutans ^2^	Sweden	0.86
25120	SV 40085/74	nutans ^2^	Sweden	0.81
25124	SV 40076/74	pallidum ^1^	Sweden	0.85
25956	Vanja	pallidum ^1^	Sweden	0.84
26127	SV 71297	pallidum ^1^	Sweden	0.81
26688	VL	pallidum ^1^	Sweden	0.81
26689	VL	pallidum ^1^	Sweden	0.85
27428	SV 73394	nutans	Sweden	0.81
27612	SV 76779	nutans ^2^ -deficiens	Sweden	0.81
27614	SV 77189	nutans ^2^ -deficiens	Sweden	0.81
27625	SV 80223	nutans ^2^ -deficiens	Sweden	0.87
27627	SV 80230	nutans ^2^ -deficiens	Sweden	0.82
27628	SV 80294	nutans ^2^ -deficiens	Sweden	0.87
27632	SV 81194	nutans ^2^ -deficiens	Sweden	0.82
27697	Taarm	nutans ^2^ -deficiens ^5^	Sweden	0.82
27712	Ola	pallidum ^1^	Sweden	0.82
30127	WW 7236	nutans ^2^	Sweden	0.87
11389	Ureiste	nutans ^2^	Finland	0.82
26191	Hja 70185	pallidum ^1^	Finland	0.85
26193	Hja 72800	pallidum ^1^	Finland	0.82
26201	J_0_ 1209	erectum ^6^	Finland	0.89
26206	J_0_ 1161	pallidum ^1^	Finland	0.89
28190	Kilta	parallelum ^4^	Finland	0.89
29264	Eero	parallelum ^4^	Finland	0.94
29295	Hja 79671	parallelum ^4^	Finland	0.96
29292	J_0_ 1370	deficiens ^5^	Finland	0.82
29293	J_0_ 1389	parallelum ^4^ pyramidatum	Finland	0.86
29297	Hja 78104	parallelum ^4^	Finland	0.83
29301	Hja 81205	nutans ^2^	Finland	0.94
28946	Hockey	nutans ^2^	Great Britain	0.85

Note: ^1^—*H. vulgare* var. *pallidum* Ser. is a synonym of *H. vulgare* L.; ^2^—*H. nutans* Alef.; ^3^—*H. vulgare* var. *medicum* Korn.; ^4^—*H. vulgare* L. convar. *vulgare* var. *parallelum* Körn.; ^5^—*H. vulgare* subsp. *deficiens* (Steud.) Á. Löve; ^6^—*H. distichon* subsp. *erectum* Schübl. et G. Martens; 4—multi-row, 2, 3, 5, and 6—two-row barley.

## Data Availability

The data presented in this study are available in article.

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
