# Peer review of "The Influence of Manganese on Growth Processes of Hordeum L. (Poaceae) Seedlings"

_plants, 2021, doi:10.3390/plants10051009_

Round 1

Reviewer 1 Report

Review in an attached file

Author Response

Answers to the first reviewer

Thanks to the referee for carefully reading the text and making comments.

The title of the work has been changed in the work.

On other notes:

The introduction has been significantly shortened. Removed some links to old works;

- Why pH = 6.5 in the control and pH = 4.0 in the experiment?

- pH 6.5 is the norm, but pH 4.0 is a model of acidic soils most typical for our country

- How many seeds were in each Petri dish?

- 50 pieces in triplicate

- There is not a single mention of Mn2+ concentration throughout the text.

- 20 mlg/L

- How many roots were measured in each experiment?

- roots were measured in at least 20 plants in triplicate

- There is no statistical processing of the results, which greatly reduces the quality of the article.

- added new tables with new data

The conclusion is also significantly changed and supplemented

Our new changes in the text are highlighted in color.

Reviewer 2 Report

The issue of how manganese affects plant function is an interesting one, especially if it is linked to environmental effects. The work submitted for review with the title “Manganese in plants. Pro et contra” unfortunately cannot be accepted and published. It contains major shortcomings. Moreover, it is written chaotically and untidy. In the Introduction section, the authors describe too much general information not fully adequate to the title of the manuscript.

In the next paragraph the authors write about general direct and indirect effects of heavy metals on animal organisms. - How does this relate to plants?  Practically the first three paragraphs could be removed from the manuscript.

In the paper, the authors use the wrong spelling of chemical compounds: oxidation degrees of anions are written as superscript, while the number of atoms of a given element in a chemical compound as subscript.

The whole manuscript should be analysed and the spelling of the units should be corrected.

In addition, English uses full stops rather than commas in decimal fractions and numbers.

The aim of the work is poorly formulated, because it is not possible to talk about the analysis of the influence of heavy metals on the example of a single element.

Besides, further explanations as to how the influence of manganese was analysed are not understandable either. The authors write once that they analysed the influence of Mn in soil, then that in aquatic systems?

Why did the authors use water at pH 6.5 as a control, and samples with manganese solutions (of concentration ? this was omitted) at pH 4.0?

The authors also do not state which manganese salts they used in their study.

The authors use different abbreviations for the same parameters, e.g. root length index is sometimes abbreviated as IDK (see Table 1) and sometimes as ROI (see page 5)?

References and description of the data in Tables 1-3 are both in the methodology and in the results? What do these tables refer to?

Latin names are written in italics.

What is the country Czechia? Please check the English name

The method of the literature list is not in accordance with the guidelines of the journal. this should be changed.

In addition, the authors quote quite a lot of literature from 1969, 1989, 1995, 1996, ect.

Author Response

Thanks to the referee for carefully reading the text and making comments.

The title of the work has been changed in the work.

On other notes:

The introduction has been significantly shortened. Removed some links to old works;

Why did the authors use water at pH 6.5 as a control, and samples with manganese solutions (of concentration ? this was omitted) at pH 4.0?

- pH 6.5 is the norm, but pH 4.0 is a model of acidic soils most typical for our country

The authors also do not state which manganese salts they used in their study.

            MnClâ‚‚

The authors use different abbreviations for the same parameters, e.g. root length index is sometimes abbreviated as IDK (see Table 1) and sometimes as ROI (see page 5)?

            fixed all abbreviations

The method of the literature list is not in accordance with the guidelines of the journal. this should be changed.

            fixed all

In addition, the authors quote quite a lot of literature from 1969, 1989, 1995, 1996, ect.

            fixed all

The conclusion is also significantly changed and supplemented

Our new changes in the text are highlighted in color.

Round 2

Reviewer 1 Report

Please, see attached file.

Author Response

Answers to the Reviewer 1

The title of the paper is changed:

"The influence of manganese on growth processes of (Hordeum L., Poaceae) seedlings"

On other notes:

The introduction has been significantly shortened.  Some links to old works are removed.

- Why pH = 6.5 in the control and pH = 4.0 in the experiment?

MF:  pH 6.5 is the norm, but pH 4.0 is a model of acidic soils most typical for our country.

- How many seeds were in each Petri dish?

MF:  50 pieces in triplicate.

- There is not a single mention of Mn2+ concentration throughout the text.

 MF:  20 mlg/L.

- How many roots were measured in each experiment?

 MF: roots were measured in at least 20 plants in triplicate.

- There is no statistical processing of the results, which greatly reduces the quality of the article.

MF:  added new tables with new data.

The conclusion is also significantly changed and supplemented.

The authors are thankful to the Reviewer 1 for carefully reading the text and making comments.

Reviewer 2 Report

The work looks much better after the corrections have been made. But it still has a few shortcomings:
1. In the research methodology. Please state what volume of solution was poure into the petri dish. I think not 4.5 liters. 
2. The authors use incorrect chemical compounds name. You cannot write six-aqueous manganese chloride and put in parentheses the anhydrous form of the salt (MnCl2). Should be MnCl2x6H2O. 
3. Distilled water with a pH of 6.5 was used as a control. One could be tempted to test in water with e.g. acetic acid and lower the pH to the same value as in the samples (pH=4) and see how the plant reacts.
4. In tables 1 and 2, the values +/_... were added to the shoot and root length values. Please state what this value is in the table description.

5. In table 1 there is an error. Should be MnCl2 x 6H2O, not concentration MnCl2 mg/L 6 H2O.
6. There is a Russian letter in the description of Table 3.

Author Response

Answers to Reviewer 2

The title of the paper is changed:

"The influence of manganese on growth processes of (Hordeum L., Poaceae) seedlings"

On other notes:

The introduction has been significantly shortened. Some links to old works are removed.

- Why did the authors use water at pH 6.5 as a control, and samples with manganese solutions (of concentration ? this was omitted) at pH 4.0?

MF:  pH 6.5 is the norm, but pH 4.0 is a model of acidic soils most typical for our country.

- The authors also do not state which manganese salts they used in their study.

MF:  It is MnClâ‚‚

- The authors use different abbreviations for the same parameters, e.g. root length index is sometimes abbreviated as IDK (see Table 1) and sometimes as ROI (see page 5)?

MF:    All abbreviations are fixed now

-The method of the literature list is not in accordance with the guidelines of the journal. this should be changed.

MF:    References are corrected in compliance with the Guidelines of the Journal.

-In addition, the authors quote quite a lot of literature from 1969, 1989, 1995, 1996, etc.

Answers to Reviewer 2

The title of the paper is changed:

"The influence of manganese on growth processes of (Hordeum L., Poaceae) seedlings"

On other notes:

The introduction has been significantly shortened. Some links to old works are removed.

- Why did the authors use water at pH 6.5 as a control, and samples with manganese solutions (of concentration ? this was omitted) at pH 4.0?

MF: - pH 6.5 is the norm, but pH 4.0 is a model of acidic soils most typical for our country

- The authors also do not state which manganese salts they used in their study.

MF: It is MnClâ‚‚

- The authors use different abbreviations for the same parameters, e.g. root length index is sometimes abbreviated as IDK (see Table 1) and sometimes as ROI (see page 5)?

MF:     all abbreviations are fixed now

-The method of the literature list is not in accordance with the guidelines of the journal. this should be changed.

MF:     References are corrected in compliance with the Guidelines of the Journal.

-In addition, the authors quote quite a lot of literature from 1969, 1989, 1995, 1996, etc.        

MF: we considered your remark and excluded some of the old references. 

MF: The conclusion is also significantly changed and supplemented.

The authors are thankful to the Reviewer for carefully reading the text and making comments.

Round 3

Reviewer 1 Report

My review in attached file.

Author Response

Answers to the Reviewer

Thank you for carefully reading the manuscript of the article and for the comments made. Below are our answers and comments to your remarks and tips for improving the article:

  • The authors have significantly revised the text of the article, added 2 tables. However, the lack of statistical analysis of the data makes the discussion of the results scanty, and the conclusions are still not statistically confirmed. All this reduces the scientific level of the article and causes many comments. There is a typo in the introduction Mn3+, but it should be Mn2+.

MF: Thanks for the comment, corrected.

  • Incorrect setting of experiments, pH in the control and in the experiment should be the same. It was necessary to acidify the water with HCl to pH = 4.0.

MF: The experiment to determine the growing conditions was carried out in an aquatic culture, varying the concentration of Mn2+ at pH = 4.0 (model of soils in Northwest Russia). The duration of the experiment is 8 days. In the experiments, we used cell growth tanks with a mesh bottom and plastic containers with a capacity of 4.5 liters. Control (according to the classical methods of seed germination, it is always set at pH = 6.0 and Mn2+ = 0).

  • In the introduction, the authors write about the toxic effect of increased concentrations of the Mn2+ ion on the growth of barley plants, and in table 1, the authors give the concentration of the salt MnCl2x4H2 It is necessary to recalculate all the concentrations used in the experiment for the content of the Mn2+ ion and in the table 1 insert exactly these concentrations. Table 1 shows the results of preliminary experiments, which made it possible to choose a concentration of manganese chloride of 20 mg/l. It is necessary to write about this in the "Materials and methods" section and transfer the table 1 in this section.

MF: Thanks for the comment, corrected. Tthe Table has been moved to the methodology section;  refined formula of manganese chloride – MnCl2 x 4H2O

  • Between what indicators was the correlation analysis carried out (table 1)? One can only guess that the correlation coefficients were calculated between the length of the shoot or root and the concentration of Mn2+

MF: Table 1 shows the concentration of salt, but not manganese.

2mg salt per liter = 25mg manganese per liter

5 mg salt per liter = 63 mg manganese per liter

10 mg - 125 mg per liter

15 mg - 189 mg per liter

20 mg - 253 mg per liter

  • In the "Materials and methods" section, the last paragraph can be removed, since it repeats the above method, and the last phrase from the previous paragraph can be moved after the discussion of table. 2.

MF: The part of the text that was a repetition is removed.

  • Why are varieties Moskovsky 121, Polyarniy 14 and Djugay combined into one group, while varieties Prairie, Odesskiy 100 and Donetskiy 8 are combined into another group? Which highlighted groups described above do they belong to? It is necessary to provide explanations. The data presented in table 3, is these the authors’ own results or literature data? Why is it only in the "Conclusion" section that the experiments were carried out on 385 samples? You need to write about this in the "Materials and methods" section.

MF: The phrase was inserted into the text of the article: To assess the resistance of barley varieties to manganese, the VIR seed bank was used, the resistance or sensitivity to manganese was tested on 385 varieties.

  • Two groups of grades were formed in relation to their metallotoxicity, in particular in relation to aluminum. The first group of varieties is resistant, the second is susceptible.The data was collected personally by the authors of the article. The article completely lacks a statistical analysis of the data, there is only a statement of facts and unfounded conclusions of the authors, not confirmed by a statistical analysis of the data. What is the reason for the high degree of RLI variability? How the data is related in the table 3 and table 4 with subspecies, varieties, identified groups, ecological and geographical origin? For the entire sample, it is necessary to carry out multivariate analysis of variance and cluster data, using the method of principal components to identify the most significant factors affecting the response of seedling roots to an increased content of manganese in the medium.

MF: The reason for the variability of the RLI parameter is in the genetic variability of the length of the roots of the studied cultivars in response to the increased content of mobile manganese in the cultivation solution.

Correlation analysis was performed between the length of the sprout and the roots and manganese content. The error in the correlation coefficient was calculated using a well-known formula. Because coefficient correlation serves as an estimate of its general parameter p, the null hypothesis was tested, i.e. assumptions that in the general population p = 0. For this, tf was calculated and compared with t st at a significance level of 0.1% for k = n - 2 = 28. tf is greater than or equal to tst.

  • For Russia, Sweden and Finland, experiments were carried out, respectively, 100, 114 and 89 samples, and in table 3 shows only the mean and Index variability. You need to present a frequency histogram showing the frequency of occurrence of different RLI values. Based on such histograms, show which subspecies and varieties are included in the class with low RLI values, and which ones with high RLI values.

MF: No such analysis has been done. The aim of the work was to show the variability of the relative length of horses under the influence of manganese.

  • For what purpose are the data presented in table 4? They are no210t analyzed in any way. The phrase “From the results obtained (Tables 3 and 4) it follows that manganese-resistant varieties and samples are found in western and northern countries with a wide distribution (what?), low pH (what?) and mobile manganese (Where? Apparently, the increased concentration of mobile manganese in the soil?) is incomprehensible. What are Western countries? Western European countries?

MF: Table 4 provides a list of the samples most resistant to free manganese.

The selected samples can be used in breeding studies to create acid-resistant varieties of barley. Isolated samples often carry genes for aluminum resistance, as shown by the authors. The presence of the property of resistance to metallotoxicity in these samples dictates the need to study them in the soil enriched with heavy metals.

Among the samples studied by us, the most stable were identified, i.e. to a lesser extent reducing root growth under stress. This is the accepted methodological approach. It is difficult to conduct such an experiment in soil culture due to the unevenness of the soil cover, various plant diseases on the control and stress backgrounds.

  • Differences in RLI values in table 4 are significant? Is the resistance of the selected variety name “Hja 81205 (ROI = 0.94), Eero (0.94), Belogorskiy (0.96) and Dzhyugiai (0.93)” significantly higher than that of the rest of the barley samples given in table 4? Why is the variety "Hja 79671 (ROI = 0.96)" not listed? If these differences are significant, then this conclusion is legitimate, but without analysis of variance, this will be an unfounded statement.

MF: The selected samples can be used in breeding studies to create acid-resistant varieties of barley. Isolated samples often carry genes for aluminum resistance, as shown by the authors. The presence of the property of resistance to metallotoxicity in these samples dictates the need to study them in the soil enriched with heavy metals.
